

# Essential role of multi-element data in interpreting elevated element concentrations in areas impacted by both natural and anthropogenic influences

Marija Petrović[1], Gordana Medunić[2] and Željka Fiket[1]

[1] Division for Marine and Environmental Research, Ruđer Bošković Institute, Zagreb, Croatia
[2] Department of Geology, Faculty of Science, University of Zagreb, Zagreb, Croatia

## ABSTRACT

**Background.** This article presents a detailed analysis of a dataset consisting of 27 elements found in soils, soil eluates, and vegetables from private gardens in a region with a long history of coal mining and burning. With coal being one of the world's most significant energy sources, and previous studies highlighting elevated element levels in vegetables from this region, the objective of this study was to identify the factors that impact soil geochemistry and metal(loid) uptake in plants.

**Methods.** Total major and trace element concentrations were analyzed in soils, soil eluates and vegetables by high resolution inductively coupled plasma mass spectrometry. The vegetable samples included six species: fennel, garlic, lettuce, parsley, onion, and radicchio. Each plant was divided into roots, stems, leaves, and/or bulbs and analyzed separately. In addition, the soil pollution status, bioavailable fractions and transfer factors from soil and soil eluates to different plant parts were determined.

**Results.** The comprehensive dataset revealed that, apart from the substrate enriched with various elements (Al, As, Co, Cr, Mo, Ni, Pb, Sb, Sn, Ti, U, V, and Zn), other anthropogenic factors such as the legacy of coal mining and combustion activities, associated industries in the area, transport, and agricultural practices, also influence the elevated element concentrations (Cd, Cu, Fe, Mn, and Se) in locally grown vegetables. The transfer factors based on element concentrations in aqueous soil eluates and element bioavailable fractions confirmed to be an effective tool for evaluating metal uptake in plants, emphasizing to some extent the effects of plant species and revealing unique patterns for each pollution source within its environmental context (*e.g.*, Cd, Mo, S, and Se in this case). The study highlights the crucial importance of utilizing comprehensive datasets that encompass a multitude of factors when interpreting the impacts of element uptake in edible plants.

## INTRODUCTION

Soil is a crucial resource with multiple functions, including ecological regulation, climate regulation, and serving as the basis for biodiversity. Most importantly, it provides plants

Corresponding author
Željka Fiket, zeljka.fiket@irb.hr

with essential nutrients, water, and air necessary for photosynthesis and forming the foundation of our food web (*Kabata-Pendias & Mukherjee, 2007*). During nutrient uptake, various metals and metalloids can be taken up into root cells and transported to other plant tissues via membrane transporters for essential or beneficial nutrients. These metals and metalloids occur naturally through weathering of rocks or atmospheric deposition. However, anthropogenic activities such as mining, improper use of fertilizers and pesticides, industrial waste discharge, and traffic exhaust have disrupted the natural cycle and content of metals and metalloids in soil, causing them to accumulate to harmful levels (*Peryea, 2001*; *Guney, Onay & Copty, 2010*; *Da Rosa Couto Couto et al., 2018*; *Akoto & Anning, 2021*; *Qin et al., 2021*; *Shan et al., 2022*). While some metals and metalloids such as Fe, Co, Zn, Cu, Mn, Mo, Ni, Se are essential for maintaining various physiological functions in plants, others such as As, Cd, Pb, Hg have no established biological functions and are considered undesirable substances (*Elbana, 2022*). Regardless of their origin or role, metals and metalloids, when present in soil at elevated concentrations, can negatively affect soil function, plant growth, and consequently human health (*Edelstein & Ben-Hur, 2018*; *Vardhan, Kumar & Panda, 2019*; *Wan et al., 2022*).

One of the simplest and widely accepted methods for characterizing human exposure to metal(loid)s through the food chain is the transfer factor (TF). By characterizing the potential ability of element transfer from soil to plant, it simultaneously combines the properties of the element itself, the properties of the soil, and the type of the plant (*Cui et al., 2004*; *Sun et al., 2013*; *Yang et al., 2014*). *Wang et al. (2006)* even considered it as a constant for a particular plant species and metal and suggested that geometric mean values are a characteristic index for a plant species. However, some authors suggested that the transfer factor based on the content of available metals in the soil could better quantify the ability of plants to uptake metals due to its sensitive value (*Wang et al., 2006*; *Yang et al., 2014*; *Gan et al., 2017*).

Although the transfer factor (TF) can be calculated for each metal of interest, its variation can be significant and dependent not only on the plant species but also among different plant parts such as root, bulb, and leaf. Furthermore, the accumulation and distribution of metals and metalloids in plants are highly influenced by numerous factors, such as their content in the surrounding soil and air, bioavailability, pH, cation exchange capacity, climatic conditions, growing season, and others (*Mirecki et al., 2015*; *Ogoko, 2015*; *Akande & Ajayi, 2017*; *Cervantes-Trejo et al., 2018*).

Understanding the mechanisms involved in metal uptake in different plant tissues is further complicated by the fact that various anthropogenic sources contribute differently to the metal balance in soil. For instance, agriculture and the use of artificial fertilizers can increase the concentration of Cu, Zn, and P in soil (*Kim & Li, 2016*; *Da Rosa Couto Couto et al., 2018*; *Elbana, 2022*). The proximity of heavy industry and steel mills can lead to elevated levels of As, Cr, Cu, Mn, Ni, Pb, Cd, Zn, and Hg in soil (*Yuan et al., 2013*; *Strezov & Chaudhary, 2017*; *Hamarashid, Fiket & Mohialdeen, 2022*), while the proximity of thermal power plants can increase the presence of various elements such as U, Se, Mo, V, Cd, Pb, and Cr (*Medunić et al., 2016*; *Radić et al., 2018*; *Singh, Shikha & Saw, 2023*). Additionally, irrigation with polluted or wastewater can also increase the concentration

of various elements depending on the composition of the water (*Cheshmazar et al., 2018*; *Yirgalem & Alemnew, 2022*). Furthermore, the literature suggests that the toxic effects may significantly change in the additive effects of multiple metals compared to the isolated toxicity of a single metal (*Kopsell & Randle, 1997*; *Bittner, 2014*; *Zhou et al., 2017*).

Despite extensive research, incomplete and insufficient datasets have made it difficult to understand how specific pollution sources affect the chemical composition of local plants under different environmental conditions. Moreover, studies often focus on a narrow group of elements, making it challenging to establish guidelines for a wide range of elements (*Cui et al., 2004*; *Wang et al., 2006*; *Sun et al., 2013*; *Yang et al., 2014*; *Gan et al., 2017*).

In order to improve our understanding of metal uptake and the impact of pollution sources, an extensive study related to distribution of 27 elements in soils, soil eluates and vegetables from private gardens in an area with a long history of coal mining and burning was conducted. The area was chosen because coal remains one of the primary energy sources worldwide, and thermal power plants fueled by coal are among the major polluters. Additionally, a preliminary study has revealed that the estimated daily intake (EDI) of several elements exceeds the maximum tolerable daily intake when consuming vegetables grown in this area (*Zorić, 2020*). This study, therefore, aims to determine the factors influencing element uptake in plants in an area affected by coal mining and combustion history by assessing the soil pollution levels, their bioavailable content, and the transfer of the studied elements from soil to vegetables.

## MATERIALS & METHODS

### Study area

Mining activity in the local coal mines of Istria (Eastern Croatia; Fig. 1) is considered the economic engine of this region from the beginning of the 20th century until the early 1970s. For decades, Raša coal was used for energy production in local households, but also in industry in both Croatia and Italy. Coal combustion and related industry have undoubtedly left negative impacts on the environment, evident not only in elevated levels of S, Se, Cd, and polycyclic aromatic hydrocarbons in local soil (*Medunić et al., 2016*; *Medunić et al., 2018a*; *Zorić, 2020*), but also in biota and local watercourses (*Medunić et al., 2018b*). Namely, coal from this region, i.e., Raša coal, is also known as superhigh-organic sulfur coal (SHOS), which has high sulfur content (up to 14%) and elevated concentrations of Mo, Se, V, and U (*CEN, 2002*; *Medunić et al., 2018a*; *Medunić et al., 2018b*). Details on the geochemistry and mineralogy of Raša coal and the effects of its combustion products on local soils, biota, and water systems are beyond the scope of this paper and can be found elsewhere (*Valković et al., 1984*; *White et al., 1990*; (*Sinninghe Damsté et al., 1999*); *Medunić et al., 2016*; *Medunić et al., 2018a*; *Medunić et al., 2018b*; *Fiket et al., 2018*; *Fiket et al., 2019*). In addition to the Raša region, this study also includes the region of the nearby town of Opatija. Although it is geographically located on the Istrian peninsula, it is separated from the Raša region by the natural border, Mount Učka. Since there was no industry in this area and Opatija has been known as a tourist and spa resort since the 19th century until today,

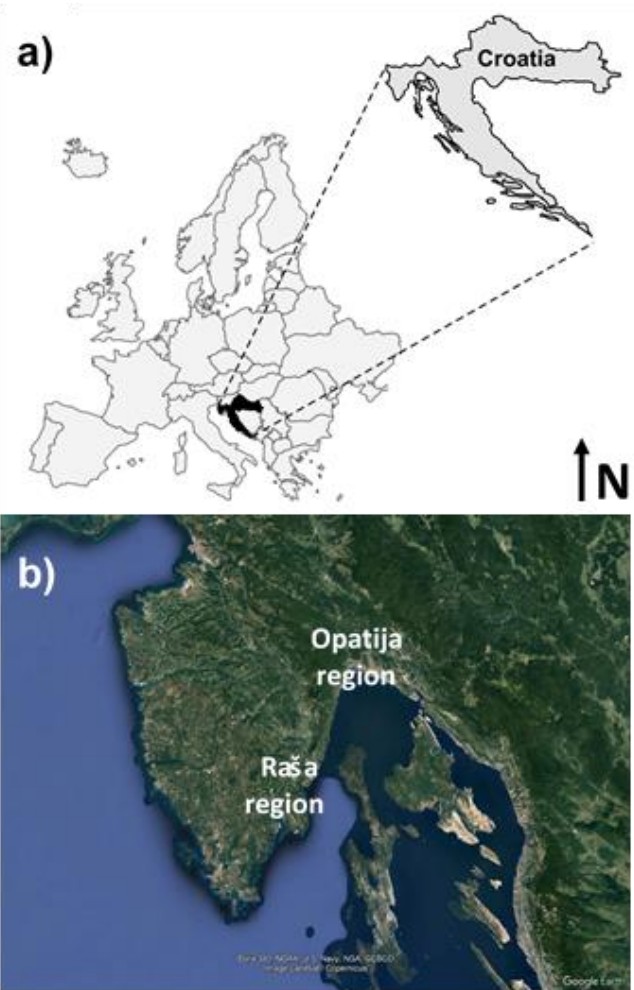

**Figure 1** Map of the study area, its geographical position (A) location of the Raša and Opatija regions (B) (Map data ©2022 Google Earth Pro).

this town is considered an area without significant anthropogenic influence, although a preliminary study (*Zorić, 2020*) suggests otherwise.

## Description of the data set

The dataset includes preliminary data (*Zorić, 2020*) on soil and plant samples collected in November 2019 at seven sites in the municipality of Raša (hereafter referred to as the Raša region), including five sites in the village of Krapan (K1-K5) and two sites in the town of Raša (R1, R2), as well as four sites in Opatija (O1-O4; hereafter referred to as the Opatija region). A total of 20 plants and 12 soil samples were collected and analyzed for total element content. Plant samples include six vegetable species, including fennel (two), garlic (two) lettuce (seven), parsley (three), onion (three), and radicchio (three). Plant parts were separated for further analysis, including roots (20), stems (three), leaves (20), and bulbs (five), resulting in a total of 48 samples. All sampling sites except one in Krapan

(site K5) were private gardens where residents grew vegetables for their own consumption. Verbal permission was obtained from the private garden owners before collecting soil and plant samples from their place.

The details of sampling and sample preparation are presented elsewhere (*Zorić, 2020*), while the data obtained are given in Tables S1 and S2 (Supplementary Material).

Briefly, the samples preparations consisted of the following procedure previously described in *Fiket & Medunić (2023)*. Soil samples were air dried, sieved through a 2 mm sieve to remove the gravel fraction, and stored until further analysis. Plants samples were washed with tap and Milli-Q water, and differentiate into different parts (roots, stems leafs and bulbs), air-dried, homogenised in an agate mill and stores until further analysis.

For total element analysis, plant sub-samples (∼0.07 g) were subjected to total digestion in the microwave oven (Multiwave 3000, Anton Paar, Graz, Austria) in a one-step procedure consisting of digestion with a mixture of 6 mL nitric acid ($HNO_3$) and 0.1 mL hydrofluoric acid (HF). Soil sub-samples (∼0.05 g) were subjected to total digestion in the microwave oven in a two-step procedure consisting of digestion with a mixture of 4 mL nitric acid ($HNO_3$, 65%, pro analysi, Kemika, Zagreb, Croatia), 1 mL hydrochloric acid (HCl) and 1 mL hydrofluoric acid (HF, 48%, pro analysi, Kemika, Zagreb, Croatia), followed by addition of 6 mL of boric acid ($H_3BO_3$, Fluka, Steinheim, Switzerland).

Prior to analysis, soil digests were 10-fold diluted, acidified with 2% (v/v) $HNO_3$ (65%, supra pur, Fluka, Steinheim, Switzerland) and In (1 µg/L) as internal standard was added. Plant digests were only acidified with 2% (v/v) $HNO_3$ (65%, supra pur, Fluka, Steinheim, Switzerland) without further dilution and In (1 µg/L) as internal standard was added.

In addition, soil pH and soil bioavailable fraction were determined for all the above elements, as described in more detail below.

## Determination of soil pH and bioavailable fraction

Soil pH was determined in a 1:10 (w/v) water suspension, *i.e.*, by adding 5 g of the soil sample to 50 ml of deionized water using a pH meter. Before measurement, the pH meter was calibrated with standard buffer solutions with pH values of 4.0, 7.0 and 10.0.

To evaluate the natural mobility of the studied elements and their bioavailable fraction in the soil, a water extraction was performed using the European Standard Batch Leaching Test (EN 12457-2, *CEN, 2002*). Milli-Q (MQ) water with a liquid-to-solid ratio of 10:1 (10 mL MQ, 1 g sample) served as the extraction medium. The prepared samples were shaken on a horizontal shaker for 24 h. After completion of the extraction time, the leachates were separated from the solids by filtration through 0.45-µm filters and acidified with 1% (v/v) $HNO_3$ (supra pur, Fluka, Steinheim, Switzerland).

Prior to analysis, samples were diluted 100-fold, acidified with 2% (v/v) $HNO_3$ (65%, supra pure, Fluka, Steinheim, Switzerland), and In (1 µg/L) was added as an internal standard.

## Multielement analysis

Multielement analysis of the prepared soil and vegetable samples (*Zorić, 2020*) was performed by high resolution inductively coupled plasma-mass spectrometry (HR-ICP-MS) using an Element 2 instrument (Thermo Fisher Scientific, Waltham, MA, USA).
Multielement analysis of the prepared soil eluates was performed by an inductively coupled plasma triple quadrupole mass spectrometer (model 8900, Agilent, USA). Typical instrument conditions and measurement parameters used throughout the work are listed in Table S3.

Mass calibration of both instruments was performed using a multielement solution (Merck KGaA, Darmstadt, Germany) containing the following elements: B, Ba, Co, Fe, Ga, In, K, Li, Lu, In, Rh, Sc, Tl, U and Y. Calibration curves were generated by external standardization with a series of standard solutions, including a blank sample. Separate standard solutions were prepared for the quantification of selected elements as follows. The standard solutions for trace element determination were prepared by appropriate dilution of a multi-element reference solution (100 ± 0.2 mg/L, Analytika, Czech Republic) containing Al, As, Ba, Cd, Co, Cr, Cs, Cu, Fe, Li, Mn, Mo, Ni, Pb, Sr, Ti, Tl, V and Zn in which single element standard solutions of U (1.000 ± 0.002 g/L, Aldrich, Milwaukee, WI, USA), Sb (1.000 ± 0.002 g/L, Analytika, Czech Republic) and Sn (1.000 ± 0.002 g/L, Analytika, Czech Republic) were added. For the determination of major elements, standard solution was prepared from single element standard solutions of 1,000 ± 2 mg/L (Analytika, Czech Republic) of K, Mg, and Ca, while for P and S, single standards were prepared from reference solutions (Analytika, Czech Republic) containing 1,000 ± 2 mg/L of these elements.

All samples were analyzed for a total concentration of 27 elements (Al, As, Ba, Ca, Cd, Co, Cr, Cu, Fe, Li, Mg, Mn, Mo, Ni, P, Pb, S, Sb, Sc, Se, Sn, Sr, Ti, Tl, U, V, and Zn).

Quality control of the analytical procedure was performed by simultaneous analysis of the blank sample and certified reference material for water (SLRS-4, NRC, Ottawa, ON, Canada). Based on the procedural blank values, detection limit values (LOD) were calculated as three times the standard deviation of five consecutive measurements of the analyte concentration and multiplied by the dilution factor (Table S4). In certified reference material, good agreement was obtained between the analyzed and certified concentrations within the analytical uncertainties (<10%) (Table S4).

## The assessment of soil pollution

Geoaccumulation indices ($I_{geo}$) and enrichment factors (EF) were calculated to quantify the extent of potential contamination of soils with metals and metalloids.

For background values, mean values for Al and other elements (for the Istrian peninsula) were taken from the Geochemical Atlas of Europe *Salminen et al. (2005)* for topsoils.

Geoaccumulation indices were calculated as follows:

$$I_{geo} = log_2 \frac{C_n}{1.5 \cdot B_n} \tag{1}$$

where $C_n$ is the measured total concentration of an element $n$ in a soil sample and $B_n$ is the background geochemical concentration of the same element in the soil, given by *Salminen et al. (2005)*. There are six classes of $I_{geo}$ (*Müller, 1981*): practically uncontaminated ($I_{geo} \leq 0$), uncontaminated to moderately contaminated ($0 < I_{geo} < 1$), moderately contaminated ($1 < I_{geo} < 2$), moderately to heavily contaminated ($2 < I_{geo} < 3$), heavily contaminated ($3$

$<I_{geo} < 4$), heavily to extremely contaminated ($4 < I_{geo} < 5$), and extremely contaminated ($5 < I_{geo}$).

Enrichment factors were calculated as follows:

$$EF = \frac{(C_n/C_{Al})_{sample}}{(B_n/B_{Al})_{background}} \tag{2}$$

where $C_n$ and $C_{Al}$ (sample) are the total concentrations of an investigated element $n$ and a reference element, respectively, in a soil sample, while $B_n$ and $B_{Al}$ (background) are the concentrations of an investigated element n and a reference element, respectively, reported for the wider study area (*Salminen et al., 2005*). The contamination categories (*Loska, Wiechula & Korus, 2004*) are as follows: insufficient to minimal enrichment (EF<2), moderate enrichment (EF = 2–5), considerable enrichment (EF = 5–20), very high enrichment (EF = 20–40), and extremely high enrichment (EF>40).

### Transfer factors

The accumulation of metals and metalloids from soil into vegetables was evaluated using the transfer factor (TF) and adjusted transfer factor (TF$_{adj.}$) calculated as follows:

$$TF = \frac{(C_n)_{plant}}{(C_n)_{soil}} \tag{3}$$

$$TF_{adj.} = \frac{(C_n)_{plant}}{(C_n)_{soil\ eluate}} \tag{4}$$

where $C_n$ are the concentrations of a studied element $n$ in plant and soil or soil eluates at the same site.

### Statistical analysis

Data were statistically analysed using STATISTICA 7.0 (StatSoft, Inc, Tulsa, OK, USA). Correlation analysis was used to determine element association via Pearson correlations between soils and plants. Differences between groups (soils, plant parts and plant species) were tested using Kruskal-Wallis analysis of variance (ANOVA), followed by pairwise comparison using Dunn's method, with significance level set at $p < 0.05$. Multivariate principal component analysis (PCA) was performed with the data matrix consisting of transfer factors.

## RESULTS

### Soil pH and element data

The soil pH ranges from 6.9 to 9.0 with an average value of 8.4, suggesting a predominantly alkaline reaction of the studied soils. Soil pH values and element data are given in the Supplementary Material S1, while Table 1 shows the element distribution in the soils of the two regions studied (Raša and Opatija), including the minimum, maximum and average values of *Zorić (2020)*. The same table also shows their comparison with literature values for soils from this region and Raša coal ash landfill (*Petrović et al., 2022*), average values for medium-textured soils (*Kabata-Pendias & Mukherjee, 2007*) and existing Croatian

**Table 1  Distribution of elements (expressed in mg/kg or \*g/kg) in the soils of the Raša and Opatija region reported in preliminary study (*Zorić, 2020*) and their comparison with the literature values.**

|  | Raša region[1] | | | Opatija region[1] | | | Soil in Raša region[2] | Raša coal ash landfill[2] | Medium-textured soils[3] | World silty and loamy soils[4] |
|---|---|---|---|---|---|---|---|---|---|---|
|  | min | max | avg | min | max | avg |  |  |  |  |
| Al* | 20.0 | 77.2 | 50.9 | 51.4 | 94.1 | 73.7 | 69.3 | 17.4 |  |  |
| As | 6.7 | 23.2 | 14.8 | 17.4 | 35.7 | 25.1 | 14.8 | 7.26 |  | 1.3–27 |
| Ba | 277 | 102 | 458 | 244 | 393 | 304 | 381 | 98.9 |  |  |
| Ca* | 40.0 | 216 | 127 | 2.3 | 3.6 | 53.4 |  |  |  |  |
| Cd | 1.0 | 4.1 | 1.66 | 0.9 | 1.3 | 1.13 | 0.83 | 0.25 | 0.5–1.0 | 0.08–1.61 |
| Co | 5.9 | 20.2 | 14.5 | 14.1 | 27.7 | 20.8 | 41.5 | 5.12 |  |  |
| Cr | 47.6 | 161 | 110 | 101 | 172 | 136 | 167 | 86.8 | 40–80 | 4–1100 |
| Cu | 50.0 | 85.3 | 62.7 | 61.3 | 85.9 | 72.4 | 37.8 | 40.8 | 60–90 | 4–100 |
| Fe* | 18.5 | 48.5 | 32.3 | 31.2 | 55.3 | 43.3 | 41.5 | 12.9 |  |  |
| Li | 14.7 | 63.9 | 39.7 | 76.9 | 202 | 119 | 53.8 | 26.6 |  |  |
| Mg* | 6.6 | 8.8 | 7.81 | 5.8 | 10.2 | 80.1 |  |  |  |  |
| Mn* | 466 | 2339 | 1.12 | 961 | 1427 | 1.24 | 2.26 | 0.08 |  | 0.045–9.2 |
| Mo | 3.2 | 17.7 | 7.29 | 2.0 | 5.2 | 3.33 | 2.89 | 18.4 |  | 0.1–7.2 |
| Ni | 60.4 | 115 | 90.3 | 90.9 | 137 | 109 | 96.3 | 42.7 | 30–50 | 3–110 |
| P* | 0.9 | 4.4 | 2.03 | 1.8 | 2.2 | 1.94 | 0.66 | 0.09 |  |  |
| Pb | 37.7 | 408 | 114 | 43.5 | 80.3 | 60.6 | 48.2 | 3.98 | 50–100 | 1.5–70 |
| S | 1.0 | 5.0 | 2.91 | 0.4 | 0.9 | 0.57 | 0.2 | 2.46 |  |  |
| Sb | 1.2 | 5.6 | 2.25 | 1.7 | 2.6 | 2.09 | 1.64 | 0.40 |  |  |
| Sc | 3.7 | 14.2 | 9.30 | 9.4 | 17.8 | 13.6 | 16.1 | 4.85 |  |  |
| Se | 1.9 | 7.3 | 4.6 | 0.1 | 4.6 | 2.10 | 21.8 | 22.9 |  | 0.02–1.9 |
| Sn | 5.0 | 9.7 | 6.84 | 5.1 | 7.2 | 6.57 | 4.7 | 1.62 |  |  |
| Sr | 115 | 352 | 235 | 84.8 | 121 | 108 | 128 | 1013 |  | 15–1000 |
| Ti | 1.2 | 4.9 | 3.19 | 3.2 | 5.9 | 4.55 | 6.65 | 1.14 |  |  |
| Tl | 0.3 | 1.3 | 0.77 | 1.3 | 2.6 | 1.93 | 1.04 | 0.15 |  |  |
| U | 1.7 | 5.2 | 3.68 | 3.5 | 6.2 | 4.99 | 4.86 | 22.2 |  |  |
| V | 55.4 | 193 | 129 | 131 | 227 | 187 | 183 | 175 |  | 15–330 |
| Zn | 144 | 797 | 348 | 192 | 442 | 289 | 120 | 17.8 |  |  |

**Notes.**

[1] *Zorić (2020)*.

[2] *Petrović et al. (2022)*.

[3] Croatian legislative values for medium-textured soils (silty and loamy) (NN 32/2010).

[4] Ranges of total concentrations of trace elements in silty and loamy soils calculated on the world scale (*Kabata-Pendias & Mukherjee, 2007*).

legislation for this type of soils (*NN 32, 2010*). It can be seen that the values given for most elements are comparable with the literature values for soils in the region (*Petrović et al., 2022*). However, compared to the global average for medium-textured soils (*Kabata-Pendias & Mukherjee, 2007*) and to the maximum permissible values according to the Croatian legislation (*NN 32, 2010*), they are characterised by increased values for Cd, Cr, Mo, Ni, Pb, and Se (Table 1).

It should be emphasised that the soils in the Raša and Opatija region, formed as residual soils over carbonate rocks, are naturally enriched with Al, As, Cd, Co, Cr, Mo, Ni, Pb, Sb, Sn, Ti, U, V and Zn compared to the European soils *Salminen et al. (2005)*.

**Table 2 Results of the Mann-Whitney U test for comparing elements content in soil between two regions. Bold tests are significant at $p < 0.050$.**

|  | Rank Sum Group 1 | Rank Sum Group 2 | U | Z | p-level |
|---|---|---|---|---|---|
| **Al** | 34.0 | 44.0 | 6.0 | −1.87 | 0.06 |
| **As** | **32.0** | **46.0** | **4.0** | **−2.19** | **0.03** |
| **Ba** | 57.0 | 21.0 | 6.0 | 1.87 | 0.06 |
| **Ca** | 56.0 | 22.0 | 7.0 | 1.71 | 0.09 |
| **Cd** | 52.0 | 26.0 | 11.0 | 1.06 | 0.29 |
| **Co** | 34.0 | 44.0 | 6.0 | −1.87 | 0.06 |
| **Cr** | 40.0 | 38.0 | 12.0 | −0.89 | 0.37 |
| **Cs** | **33.0** | **45.0** | **5.0** | **−2.03** | **0.04** |
| **Cu** | 36.0 | 42.0 | 8.0 | −1.54 | 0.12 |
| **Fe** | 36.0 | 42.0 | 8.0 | −1.54 | 0.12 |
| **Li** | **28.0** | **50.0** | **0.0** | **−2.84** | **0.00** |
| **Mg** | 44.0 | 34.0 | 16.0 | −0.24 | 0.81 |
| **Mn** | 39.0 | 39.0 | 11.0 | −1.06 | 0.29 |
| **Mo** | 56.0 | 22.0 | 7.0 | 1.71 | 0.09 |
| **Ni** | 38.0 | 40.0 | 10.0 | −1.22 | 0.22 |
| **P** | 43.0 | 35.0 | 15.0 | −0.41 | 0.68 |
| **Pb** | 47.0 | 31.0 | 16.0 | 0.24 | 0.81 |
| **S** | **63.0** | **15.0** | **0.0** | **2.84** | **0.00** |
| **Sb** | 41.0 | 37.0 | 13.0 | −0.73 | 0.46 |
| **Sc** | 34.0 | 44.0 | 6.0 | −1.87 | 0.06 |
| **Se** | 57.0 | 21.0 | 6.0 | 1.87 | 0.06 |
| **Sn** | 46.0 | 32.0 | 17.0 | 0.08 | 0.94 |
| **Sr** | **60.0** | **18.0** | **3.0** | **2.35** | **0.02** |
| **Ti** | 34.0 | 44.0 | 6.0 | −1.87 | 0.06 |
| **Tl** | **28.0** | **50.0** | **0.0** | **−2.84** | **0.00** |
| **U** | 36.0 | 42.0 | 8.0 | −1.54 | 0.12 |
| **V** | 34.0 | 44.0 | 6.0 | −1.87 | 0.06 |
| **Zn** | 44.0 | 34.0 | 16.0 | −0.24 | 0.81 |

**Notes.**
Group 1: Raša region ($N = 7$).
Group 2: Opatija region ($N = 5$).

## Bioavailable fraction in soil

The concentrations of the elements in soil eluates and the calculated bioavailable fractions (%) are shown in Table 2. Since Sc and Tl in soil eluates were below the detection limit, they were not considered in further discussion.

On average, concentrations of less than 0.001 mg/kg for Cd, Co, Cr, Li, Pb, Sb; from 0.001 to 0.01 mg/kg for As, Ba, Cu, Mo, Ni, Se, Sn, U, and V; from 0.01 to 0.1 mg/kg for Fe, Mn, Sr, and Ti; and >0.1 mg/kg for Ca, Mg, P, S, and Zn were found in soil eluates.

The bioavailable fractions varied by up to four orders of magnitude and were as high as 3.7% (S) with an average of 0.1%. In addition, the proportions of bioavailable fractions per element varied by up to three orders of magnitude and had the largest differences between
samples for Al, Ba, Ti, and Zn. On average, the lowest available fractions (<0.005%) were observed for Al, Ba, Co, Cr, Fe, Li, Mn, Ni, Pb and Ti, slightly higher (0.005−0.05%) for As, Ca, Cd, Cu, Mg, Sb, Sn, Sr, and V, even higher (0.05−0.5%) for Mo, P, Se, U, and Zn, and the highest (>0.5%) for S.

For most elements, the reported concentrations are lower than or comparable to the minimum values reported by *Fiket et al. (2019)* for eluates from regional soils contaminated by coal mining and associated industries. The only exceptions are Sn, Ti, and Zn, for which the values in some eluates are comparable to the highest values reported by these authors.

Accordingly, most elements exhibited low mobility, *i.e.*, bioavailability, with the exception of nutrients (P, S, and Zn) and elements that tend to form oxyanions (Mo, Se, and U). The latter is consistent with an average soil pH in the alkaline range (8.4) and the presence of carbonates, which increase the retention of heavy metals at such conditions (*Elbana, 2022*). At the same time, such pH favours the mobility of elements present as oxyanions and, in this particular case, elements whose presence in soils is subject to moderate to high contamination by anthropogenic influences, such as Mo, Se, and U (*Brent et al., 2005*). And although the observed mobility is generally low and consistent with previous data (*Fiket et al., 2019*), cytotoxic effects of soils have been reported even at bioavailable levels as low as 0.2% (*Fiket et al., 2019*).

## DISCUSSION

### Soil quality evaluation

In order to adequately assess the soil quality at the studied sites and to identify possible sources of pollution, enrichment factors and geoaccumulation indices were calculated for all elements, using the upper values given by *Salminen et al. (2005)* for the top soils of the region as a reference. The only exceptions were Li and Se, for which no data are available in the Geochemical Atlas of Europe and which were not included in Fig. 2 (Li and Se) and Fig. 3 (Li).

From the $I_{geo}$ results (Fig. 2), it can be concluded that the soil at all sites contains the expected levels of geogenic elements such as Al, Sc, and Ti. The same is true for typical heavy metals and metalloids such as As, Co, Cr, Fe, Mn, Ni, and Tl, which are usually associated with various anthropogenic activities, mainly industrial (*Kierczak et al., 2008*; *Sofilić et al., 2013*). The only exception is site O5 in Opatija, where the values of As, Co, Cr, and Ni describe the soil as uncontaminated to moderately contaminated. For U, V and Se, elements present in high proportions in the Raša coal (*Medunić et al., 2016*; *Medunić et al., 2018b*), most sites can be described as unpolluted, with the exception of one site in Raša (R2) and three sites in Opatija (O3-O5), where the soils are uncontaminated to moderately contaminated for U. However, this is not the case for S and Mo (Fig. 2), the other two elements usually found enriched in Raša coal, for which the studied sites range from uncontaminated to moderately to highly contaminated for Mo and uncontaminated to highly contaminated for S, with higher contamination levels found in the Raša region than in the Opatija region.

Both investigated areas also have elevated values for Pb, Sb and Sn, as well as for Cu, Zn and P, which are characteristic of uncontaminated to moderately to heavily contaminated

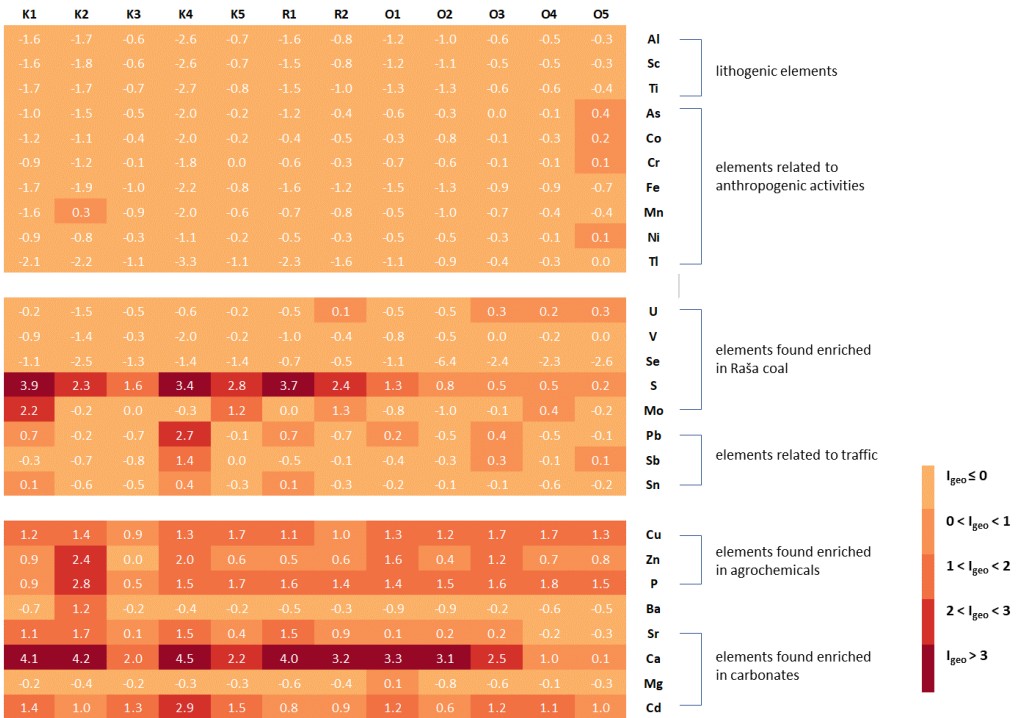

**Figure 2** Geoaccumulation indices (I_geo) in studied soils in relation to groups of elements associated with certain natural or anthropogenic sources.

soils and are only slightly higher in the Raša area. While the first group of elements (Pb, Sb and Sn) is usually associated with transport (*Alloway, 2013*; *Clemente, 2013*), the latter (Cu, Zn and P) is significantly elevated in agrochemicals (*Kim & Li, 2016*; *Da Rosa Couto Couto et al., 2018*; *Elbana, 2022*). Strong enrichment of Ca, Sr, Mg, and Cd, elements associated with the carbonate fraction in the eastern Adriatic *Salminen et al. (2005)*, was also found in the studied soils. This is not surprising considering that the preparation of the samples in the preliminary study by *Zorić (2020)* was not the same as the procedure to which the samples were subjected in the study by *Salminen et al. (2005)*. Namely, in the mentioned preliminary study *Zorić (2020)*, the soil samples were sieved through a 2-mm sieve, while *Salminen et al. (2005)* studied soil samples sieved through a 63-um sieve that removes carbonates.

In addition, elevated Cd, Cu, Pb, and Zn levels in soil are also typical for sites contaminated by mining and metal industry (*Kierczak et al., 2008*; *Sofilić et al., 2013*), while elevated Cd levels were also found in soils near the nearby thermal power plant (*Medunić et al., 2016*). The origin of these elements in the Raša and Opatija region can therefore be attributed to several sources, including the geological and pedological background, the legacy of decades of coal mining and combustion activities, the associated industries in the area, and the transport and agricultural practices used by the local population in their vegetable gardens.
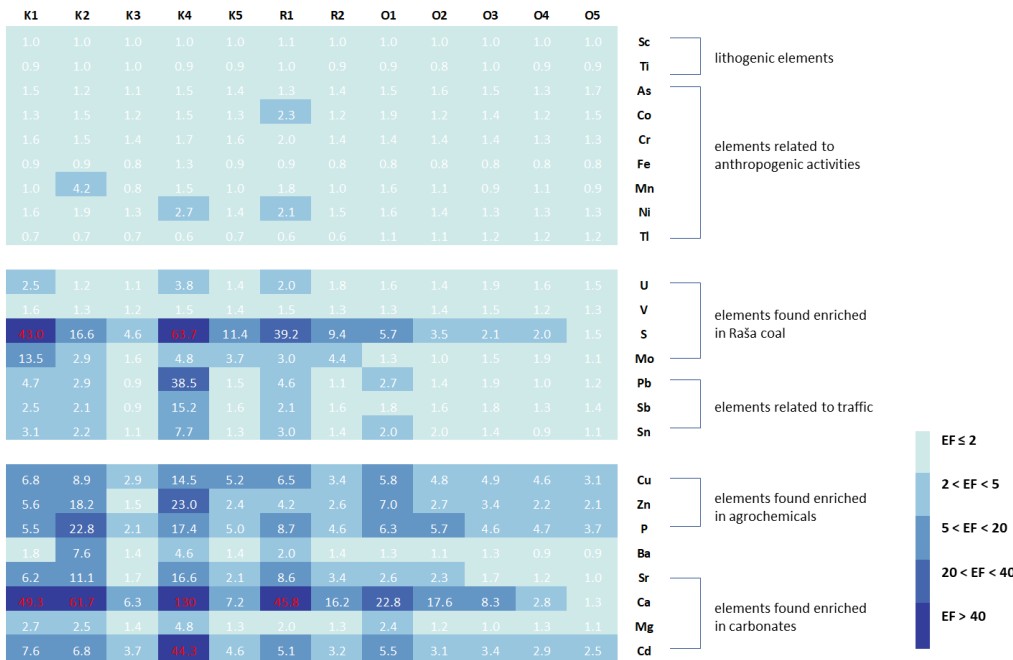

**Figure 3** Enrichment factors (EF) in studied soils in relation to groups of elements associated with certain natural or anthropogenic sources.

The enrichment factors (Fig. 3) are consistent with the geoaccumulation indices and indicate moderate enrichment of S in the Opatija area and considerable to extremely high enrichment in the Raša area, accompanied by moderate to considerable enrichment of Mo. Soils also showed minimal to moderate enrichment for Pb, Sb, and Sn in the Opatija area and minimal to extremely high enrichment in the Raša area. Enrichment with elements characteristic of agrochemicals is observed throughout the region, with slightly higher enrichment in the Raša area, reaching very high enrichment for Zn and P in some places. The elements of the carbonate fraction also show significant enrichment, but this is due to the sample preparation itself and not to the actual enrichment. Other elements show insufficient to minimal enrichment and reflect background values.

Despite the observed differences in soil enrichment for some element groups, a statistically significant difference ($p < 0.05$; Table 3) between the soils of the two regions studied was found only for As, Li, S, Sr, and Tl. For As, Li and Tl, the soil samples from the Opatija region had higher values for these elements, while the soils from the Raša region had higher values for S and Sr.

It can be concluded that the chemical composition of soils in the Raša and Opatija regions is influenced by both natural and anthropogenic factors. These include the geological and pedological background enriched with various elements, local industry and traffic, the use of agrochemicals and, above all, the long history of coal mining and burning in Raša, which is more pronounced in the Raša region.

Petrović et al. (2023), *PeerJ*, DOI 10.7717/peerj.15904

**Table 3 Element concentrations in eluates (in mg/L) and their bioavailable soil fractions (in %).** Bolded in black are bioavailable fractions > 0.05%.

| | Al | As | Ba | Ca | Cd | Co | Cr | Cu | Fe | Li | Mg | Mn | Mo |
|---|---|---|---|---|---|---|---|---|---|---|---|---|---|
| **Element concentrations in eluates (mg/L)** | | | | | | | | | | | | | |
| K1 | 0.036 | 0.002 | 0.020 | 19.4 | 0.0006 | 0.0002 | 0.0003 | 0.013 | 0.020 | 0.0009 | 0.87 | 0.014 | 0.005 |
| K2 | 0.134 | 0.002 | 0.032 | 31.7 | 0.0006 | 0.0002 | 0.0007 | 0.019 | 0.142 | 0.0012 | 2.65 | 0.050 | 0.004 |
| K3 | 0.030 | 0.001 | <0.001 | 12.0 | 0.0006 | <0.0001 | <0.0001 | 0.002 | 0.021 | 0.0008 | 0.62 | 0.031 | 0.007 |
| K4 | 0.085 | 0.001 | 0.005 | 15.1 | 0.0006 | <0.0001 | 0.0003 | 0.009 | 0.039 | 0.0012 | 0.72 | 0.014 | 0.004 |
| K5 | 0.037 | 0.003 | 0.004 | 15.1 | 0.0005 | <0.0001 | 0.0001 | 0.010 | 0.026 | 0.0008 | 0.51 | 0.012 | 0.003 |
| R1 | 0.051 | 0.002 | 0.002 | 11.0 | 0.0006 | <0.0001 | 0.0002 | 0.009 | 0.022 | 0.0011 | 0.84 | 0.010 | 0.003 |
| R2 | 0.035 | 0.003 | 0.003 | 13.9 | 0.0005 | <0.0001 | 0.0001 | 0.001 | 0.015 | 0.0008 | 0.79 | 0.005 | 0.004 |
| O1 | 0.176 | 0.001 | 0.013 | 25.5 | 0.0006 | 0.0004 | 0.0004 | 0.012 | 0.056 | 0.0008 | 1.13 | 0.048 | 0.003 |
| O2 | 0.140 | 0.001 | 0.006 | 16.7 | 0.0005 | <0.0001 | 0.0004 | 0.010 | 0.123 | 0.0006 | 1.82 | 0.019 | 0.003 |
| O3 | 0.030 | 0.003 | 0.018 | 16.2 | 0.0005 | 0.0003 | <0.0001 | 0.009 | 0.014 | 0.0008 | 0.76 | 0.030 | 0.004 |
| O4 | 0.041 | 0.001 | <0.001 | 13.1 | 0.0006 | <0.0001 | <0.0001 | 0.010 | 0.023 | 0.0008 | 0.98 | 0.012 | 0.003 |
| O5 | 2.619 | 0.001 | 0.005 | 11.3 | 0.0005 | 0.0002 | <0.0001 | 0.004 | 0.047 | 0.0009 | 0.77 | 0.032 | 0.003 |
| **Bioavailable fraction (%)** | | | | | | | | | | | | | |
| K1 | <0.0001 | 0.015 | 0.0074 | 0.012 | 0.042 | 0.002 | 0.0003 | 0.021 | <0.0001 | 0.0028 | 0.010 | 0.002 | 0.028 |
| K2 | 0.0004 | 0.017 | 0.0030 | 0.017 | 0.049 | 0.002 | 0.0009 | 0.027 | 0.0006 | 0.0045 | 0.036 | 0.002 | **0.110** |
| K3 | <0.0001 | 0.004 | <0.0002 | 0.030 | 0.041 | <0.001 | <0.0001 | 0.004 | <0.0001 | 0.0014 | 0.007 | 0.003 | **0.184** |
| K4 | 0.0004 | 0.016 | 0.0014 | 0.007 | 0.013 | <0.001 | 0.0007 | 0.014 | 0.0002 | 0.0078 | 0.009 | 0.003 | **0.133** |
| K5 | <0.0001 | 0.015 | 0.0010 | 0.034 | 0.034 | <0.001 | 0.0001 | 0.011 | <0.0001 | 0.0013 | 0.006 | 0.001 | 0.036 |
| R1 | <0.0001 | 0.015 | 0.0005 | 0.007 | **0.059** | <0.001 | 0.0002 | 0.016 | <0.0001 | 0.0032 | 0.013 | 0.001 | **0.081** |
| R2 | <0.0001 | 0.013 | 0.0007 | 0.015 | **0.054** | <0.001 | 0.0001 | 0.002 | <0.0001 | 0.0016 | 0.011 | 0.001 | 0.043 |
| O1 | 0.0003 | 0.007 | 0.0051 | 0.026 | 0.044 | 0.002 | 0.0004 | 0.019 | 0.0002 | 0.0011 | 0.011 | 0.004 | **0.129** |
| O2 | 0.0002 | 0.007 | 0.0023 | 0.020 | **0.063** | <0.001 | 0.0004 | 0.016 | 0.0004 | 0.0007 | 0.031 | 0.002 | **0.133** |
| O3 | <0.0001 | 0.011 | 0.0045 | 0.030 | 0.042 | 0.002 | <0.0001 | 0.011 | <0.0001 | 0.0006 | 0.012 | 0.003 | **0.103** |
| O4 | <0.0001 | 0.004 | <0.0002 | **0.065** | 0.049 | <0.001 | <0.0001 | 0.012 | <0.0001 | 0.0009 | 0.010 | 0.001 | **0.053** |
| O5 | 0.0028 | 0.003 | 0.0017 | **0.109** | 0.050 | 0.002 | <0.0001 | 0.005 | <0.0001 | 0.0005 | 0.009 | 0.002 | **0.094** |

**Table 3** (*continued*)

| | Ni | P | Pb | S | Sb | Se | Sn | Sr | Ti | U | V | Zn |
|---|---|---|---|---|---|---|---|---|---|---|---|---|
| | | | | | **Element concentrations in eluates (mg/L)** | | | | | | | |
| K1 | 0.002 | 0.41 | <0.0001 | 38.4 | 0.0013 | 0.0029 | 0.0011 | 0.051 | 0.0017 | 0.0031 | 0.012 | <0.0001 |
| K2 | 0.004 | 1.13 | <0.0001 | 39.5 | 0.0009 | 0.0021 | 0.0013 | 0.069 | 0.0021 | 0.0025 | 0.009 | 0.0055 |
| K3 | 0.002 | 0.39 | <0.0001 | 38.1 | 0.0006 | 0.0014 | 0.0033 | 0.041 | 0.0012 | 0.0025 | 0.004 | <0.0001 |
| K4 | 0.002 | 0.83 | 0.0028 | 40.8 | 0.0018 | 0.0015 | 0.0010 | 0.055 | 0.0027 | 0.0025 | 0.010 | 0.1286 |
| K5 | 0.004 | 1.32 | <0.0001 | 42.3 | 0.0008 | 0.0018 | 0.0015 | 0.037 | 0.0017 | 0.0026 | 0.014 | <0.0001 |
| R1 | 0.003 | 0.74 | <0.0001 | 34.9 | 0.0011 | 0.0025 | 0.0010 | 0.039 | 0.0036 | 0.0027 | 0.012 | 0.0076 |
| R2 | 0.001 | 0.73 | <0.0001 | 36.8 | 0.0011 | 0.0028 | 0.0007 | 0.037 | 0.0018 | 0.0030 | 0.013 | <0.0001 |
| O1 | 0.003 | 0.95 | <0.0001 | 5.0 | 0.0008 | 0.0011 | 0.0067 | 0.040 | 0.0034 | 0.0026 | 0.008 | 0.0024 |
| O2 | 0.002 | 0.94 | <0.0001 | 3.9 | 0.0008 | 0.0007 | 0.0052 | 0.027 | 0.0060 | 0.0027 | 0.010 | 0.2149 |
| O3 | 0.002 | 1.43 | <0.0001 | 3.6 | 0.0008 | 0.0009 | 0.0011 | 0.028 | 0.0019 | 0.0026 | 0.012 | <0.0001 |
| O4 | 0.002 | 1.53 | <0.0001 | 4.0 | 0.0007 | 0.0010 | 0.0008 | 0.023 | 0.0010 | 0.0025 | 0.008 | <0.0001 |
| O5 | 0.004 | 0.41 | <0.0001 | 3.6 | 0.0007 | 0.0009 | 0.0038 | 0.016 | 0.1413 | 0.0025 | 0.007 | 3.843 |
| | | | | | **Bioavailable fraction (%)** | | | | | | | |
| | Ni | P | Pb | S | Sb | Se | Sn | Sr | Ti | U | V | Zn |
| K1 | 0.003 | 0.034 | <0.0001 | **0.77** | **0.070** | **0.060** | 0.014 | 0.023 | <0.0001 | **0.073** | 0.010 | <0.0001 |
| K2 | 0.005 | 0.025 | <0.0001 | **2.31** | **0.063** | **0.113** | 0.026 | 0.019 | <0.0001 | **0.144** | 0.011 | 0.0007 |
| K3 | 0.002 | 0.044 | 0.0003 | **3.74** | 0.046 | 0.033 | **0.063** | 0.035 | <0.0001 | **0.070** | 0.002 | <0.0001 |
| K4 | 0.004 | 0.043 | 0.0007 | **1.10** | 0.032 | 0.038 | 0.010 | 0.018 | 0.0002 | **0.078** | 0.017 | 0.0226 |
| K5 | 0.003 | **0.064** | 0.0002 | **1.71** | 0.039 | 0.045 | 0.025 | 0.027 | <0.0001 | **0.059** | 0.007 | <0.0001 |
| R1 | 0.003 | 0.038 | <0.0001 | **0.76** | **0.071** | 0.040 | 0.013 | 0.012 | <0.0001 | **0.078** | 0.011 | 0.0036 |
| R2 | 0.001 | 0.043 | 0.0003 | **1.99** | **0.052** | 0.038 | 0.011 | 0.018 | <0.0001 | **0.057** | 0.008 | <0.0001 |
| O1 | 0.003 | **0.053** | <0.0001 | **0.59** | 0.050 | 0.024 | **0.102** | 0.034 | <0.0001 | **0.075** | 0.006 | 0.0006 |
| O2 | 0.003 | **0.051** | <0.0001 | **0.66** | 0.044 | **0.565** | **0.072** | 0.022 | 0.0002 | **0.077** | 0.007 | **0.1117** |
| O3 | 0.002 | **0.072** | <0.0001 | **0.75** | 0.031 | 0.046 | 0.015 | 0.023 | <0.0001 | 0.042 | 0.005 | <0.0001 |
| O4 | 0.002 | **0.069** | 0.0002 | **0.80** | 0.033 | 0.048 | 0.016 | 0.024 | <0.0001 | 0.043 | 0.004 | <0.0001 |
| O5 | 0.003 | 0.040 | <0.0001 | **0.89** | 0.028 | **0.053** | **0.057** | 0.018 | 0.0024 | 0.041 | 0.003 | **1.551** |

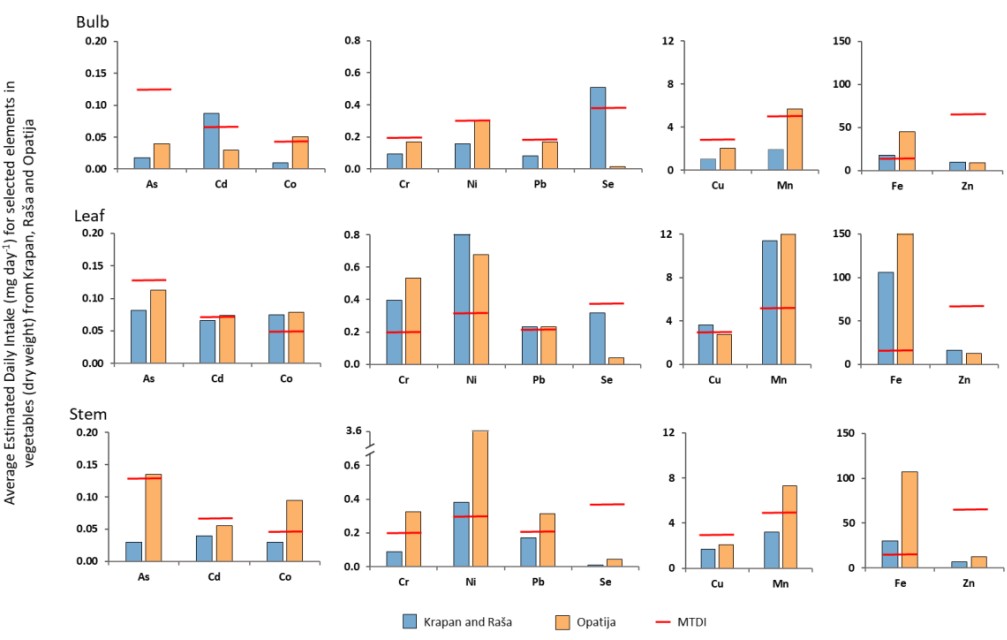

**Figure 4  Average estimated daily intake (mg day$^{-1}$) for selected elements in vegetables from studied regions compared to MTDI (mg day$^{-1}$).**

## Element levels and health risk assessment

The estimated daily intake (EDI) of eleven elements (As, Cd, Co, Cr, Cu, Fe, Mn, Ni, Pb, Se, and Zn) in different vegetables from Raša and Opatija regions reported by *Zorić (2020)* is shown in Fig. 4A. The average body weight used for the calculation was 65 kg, while 174 g (FW) was assumed as the daily dose of vegetable intake (Div; *EFSA Europa, 2011*), but converted to DW using the conversion factor 0.085 (*Rattan et al., 2005*). As can be seen from Fig. 4, the content of elements in vegetables from both regions studied, with the exception of Zn, exceeded the maximum tolerable daily intake (MTDI).

It is interesting to note that higher values than the MTDI values were found both for elements enriched in soils (Cd, Pb, Cu) and for those whose concentrations in soils correspond to natural values (As, Co, Cr, Fe, Mn, Ni). For Zn, an element found to be very highly enriched in soil, the EDI was significantly lower than the MTDI, regardless of plant part or region of origin.

Combined data from a number of countries, including Croatia, showed that cadmium concentrations in most foods ranged from 0.01 to 0.05 mg/kg, with higher concentrations found in nuts and oilseeds, mollusks, and offal (*FAO/WHO, 2004*). Estimates of average national cadmium intake ranged from 0.7 to 6.3 mg/kg body weight per week. However, for the vegetables studied the average concentrations in bulb, leaf, and stem were 0.33, 0.28, and 0.21 mg/kg, respectively, and well above the ranges reported by (*FAO/WHO, 2004*) for Cd in food.

The hazard quotients (HQs) calculated by *Zorić (2020)* for oral intake of various elements are presented in Table S5. The HQs were calculated as the ratio of the determined dose

to the reference dose: HQ = (Div) x ($C_{element}$) / $R_fD$ x BW, where Div is the daily dose of vegetable intake (kg person$^{-1}$ day$^{-1}$), $C_{element}$ is the element concentration in vegetable samples (mg kg$^{-1}$), RfD is the oral reference dose for the specific element (mg kg$^{-1}$ day$^{-1}$), and BW is the human body weight (kg). The RfD values were taken from the Integrated Risk Information System (IRIS), an environmental assessment program operated by the USEPA. Only for S and Pb, HQs could not be calculated because no data were available.

Despite the higher EDI values found for some elements, none of the HQ values exceeded 1, indicating a low chronic risk of carcinogenic effects on human health from consumption of the vegetables studied. The highest HQ values were calculated for As in vegetables from the Opatija region (up to 0.67; Table S5).

To clarify the factors affecting the levels of major and trace elements in vegetables, the distribution of elements in relation to plant parts or species, as well as the bioavailable fraction in the soil and the transfer from soil to plant, are discussed below.

## Influence of soil on the element composition of vegetables

Correlation factors were calculated to investigate the influence of soil geochemistry on the multi-element composition of vegetables. Figure 5 shows that the composition of plants in terms of individual elements reflects their occurrence in the soil, but not exclusively. In this sense, in plants three groups of elements can be clearly distinguished. The first group consists of typical geogenic elements such as Al, Li, Sc, and Ti, and As, Co, Cr, Cu, Fe, Ni, Sn, Tl, U, and V, which show a strong positive correlation with their content in soils, indicating their common origin.

However, the contents of Mo, Pb, S, and Se in the plants show only positive correlations with these elements in the soil. The same is true to a somewhat lesser extent for Sn, while Sb concentrations in the plants also show a positive correlation with Sb, Sn, and Pb concentrations in the soil in addition to the high positive correlations with the previously mentioned lithogenic group of elements. Although Cd is generally enriched in residual soils on carbonate rocks, Cd in plants has the highest correlation with Se and Mo in the soil.

It is particularly interesting that the concentrations of Mn, P and Zn in the plants show the highest positive correlation coefficients with the Zn content of the soil. High correlation with soil Zn content was also found for Ba, Ca and Mg in vegetables. In general, plant nutrients Zn, Mn, and P correlate strongly with soil carbonate content (*i.e.*, Ca and Mg), most likely due to more intensive plant amendment in soils with higher carbonate content.

The results indicate that the correlation between lithogenic elements in plants and their levels in soil underscores the direct influence of soil geochemistry on their uptake. On the other hand, the positive correlation of certain groups of elements, such as Cd, Mo, Pb, S, and Se, or Mn, P, and Zn, with only their levels in soil or specific elements like Zn for the latter group, suggests the involvement of additional factors that impact their uptake and distribution in plants.

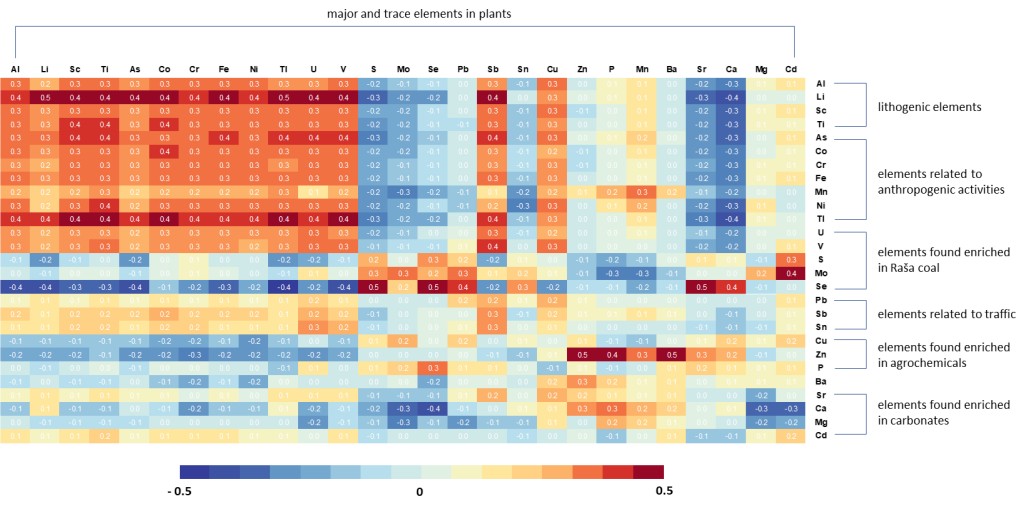

**Figure 5** **Matrix showing the correlation coefficients between element concentrations in plants and their concentrations in soils.** Intensity of shading (red for positive and blue for negative) represents the strength of the Pearson correlation coefficients.

## Influence of plant part on element uptake

To better understand the mechanisms of uptake and distribution of elements in plants, their average concentrations in the root, stem, leaf, or bulb were calculated using data from a preliminary study (*Zorić, 2020*). In the studied vegetables, the accumulation of elements was found predominantly in the root region, while for most elements it decreased toward the other organs (bulb, stem, or leaf; Fig. 6). Exceptions to this trend were Ca, Mo, P, S, and Se, whose highest values were observed in the stem (Ca), leaf (P), or bulb (Mo, S, Se). The observed decrease in metal concentrations from roots to other plant organs is consistent with general observations in the literature (*Irshad et al., 2014*; *Yabanli, Yozukmaz & Sel, 2014*; *Galić et al., 2019*). Furthermore, the study by *Sulaiman & Hamzah (2018)* on the accumulation of heavy metals in roadside plants suggests that metals are only weakly transported from the root zone to the stem, but are readily mobilized to the leaves when available in the stems, which explains their generally higher levels in leaves than in stems.

The latter is especially emphasized for phosphorus. In the literature, metabolism of this element is primarily viewed through the prism of soil fertilization, and available data on the distribution of phosphorus among plant organs confirm that it is generally more enriched in shoots than in roots. The study by *Kim & Li (2016)* on the effects of phosphorus (P) on shoot and root growth, distribution, and efficiency of phosphorus utilization in Lantana camara clearly showed that increasing the concentration of P increased the accumulation of P in all parts of the plant but mainly in the shoots, while further increasing the concentration increased the accumulation mainly in the roots and flowers. At the same time, plants accumulated a similar proportion of P in leaves and stems (70%) regardless of P availability (*Kim & Li, 2016*), which is consistent with greater P accumulation in bulbs and leaves in the observed vegetables (Fig. 6).

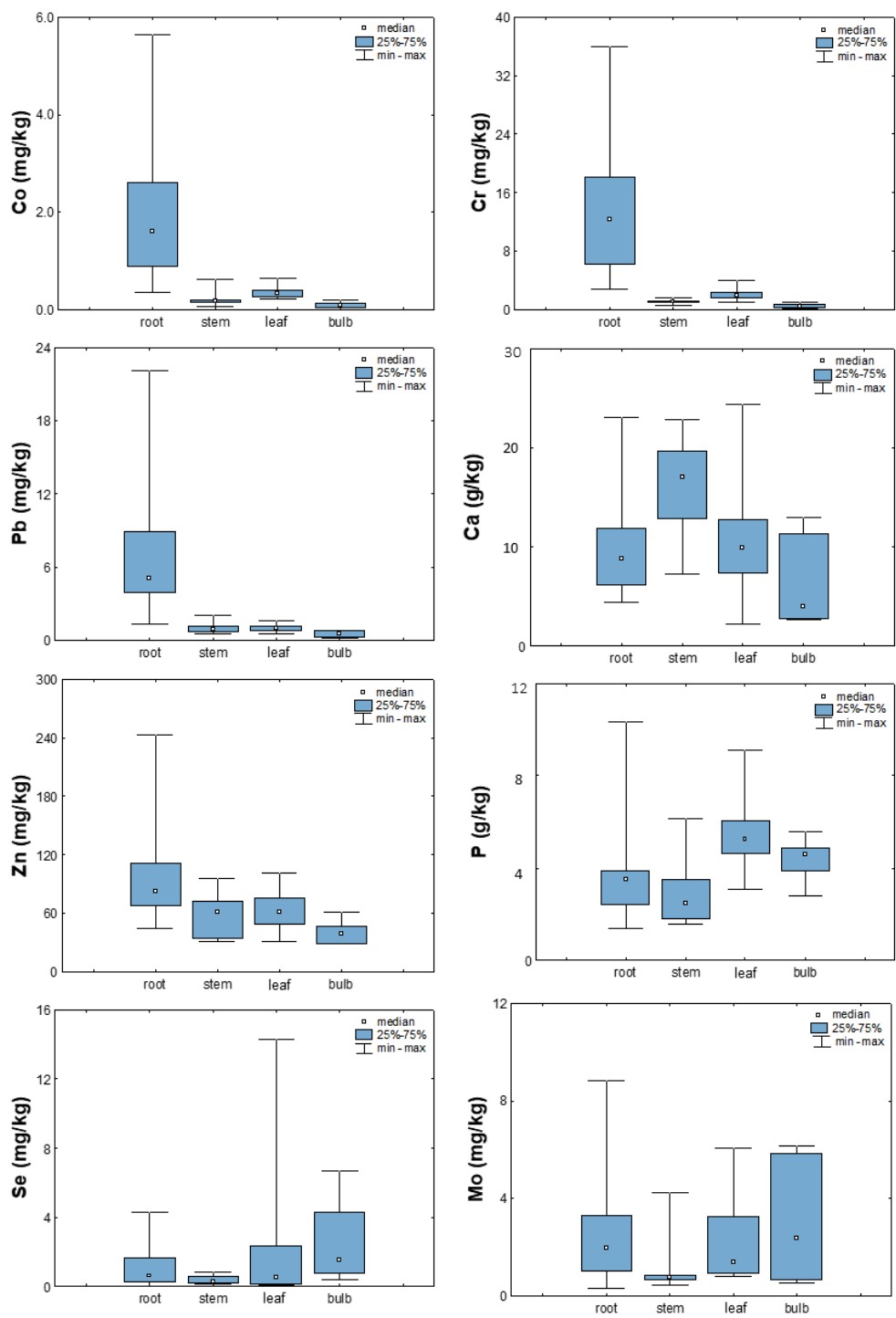

**Figure 6 Box-plots showing concentrations of selected elements in different plant parts.**

Somewhat different trends were reported for sulfur and selenium. Because of the chemical similarity of Se and S, plants and other organisms readily take up and metabolize Se through the S uptake and assimilation pathway (*Terry et al., 2000*; *Sors, Ellis & Salt, 2005*; *Pilon-Smits & Quinn, 2010*). In their study of onion cultivars, *Kopsell & Randle (1997)* observed a high positive correlation between Se concentrations and S accumulation in an onion. Onions grown in nutrient solutions supplemented with 2.0 mg $Na_2SeO_4$ per liter contained an average of 85.2 mg/kg Se and showed an average increase in S content from 5.4 to 5.9 g/kg, suggesting a synergistic relationship between selenate and S.

Furthermore, in studying molybdenum mechanisms in plants, *Bittner (2014)* concluded that the three nutrients molybdenum, iron, and sulfur appear to interact closely at different levels within a common metabolic network because sulfur is embedded in several other steps of molybdenum metabolism (*e.g.*, moco-synthesis, sulfite detoxification, molybdate uptake). The observed similarities between the distribution of S, Se and Mo among different plant parts and their highest enrichment in bulbs suggest a potential synergistic effect between these elements.

## Influence of vegetable species on the element uptake

The different possibilities of absorption and accumulation of elements in a plant depend not only on the type of element and bioavailability, pH and general soil properties (*Kabata-Pendias & Mukherjee, 2007*; *Elbana, 2022*), the part of the plant, but also on many other factors, including the plant species. For example, leafy vegetables tend to accumulate more heavy metals than root and fruiting vegetables (*Wang et al., 2006*; *Yang et al., 2014*; *Gan et al., 2017*), with the general order leafy >root >fruiting vegetables (*Ali & Al-Qahtani, 2012*; *Xu et al., 2015*; *Hu et al., 2017*).

The TFs calculated for the plant species studied here are listed in the Table S6 (Supplementary Material), while their distribution among the different plant species for selected elements is shown in Fig. 7. As can be seen from this figure, there are differences among the vegetable species studied, but it is difficult to describe them with clear trends. In general, the TF values obtained ranged from <0.001 to 3.9, with TF >1 values observed for S and P and the next highest average values for Se (0.49), Mo (0.46), and Cd (0.39). Ti (0.04), Al (0.05), and Fe (0.05) had the smallest TFs. The latter is consistent with previous data on the bioavailability of elements in the soils from the Raša region and clearly shows that the substrate, regardless of the plant species and their ability to accumulate certain elements, strongly influences the intensity of uptake of certain elements.

As mentioned earlier, transfer factors based on the concentrations of elements available in the soil are considered more sensitive to adequately quantify the uptake of elements by plants (*Wang et al., 2006*; *Yang et al., 2014*; *Gan et al., 2017*). To test this hypothesis for vegetables from the Raša and Opatija region, $TF_{adj.}$ were calculated based on the bioavailable content of the elements and are shown in Fig. 8 for Co, Cr, Mo and Se. The figure shows that the mean values of $TF_{adj.}$ differ significantly less than TF between the different vegetables, although the variability is still element-dependent and is greatest for Mo. This confirms that the adjusted transfer factors indeed give a better insight into

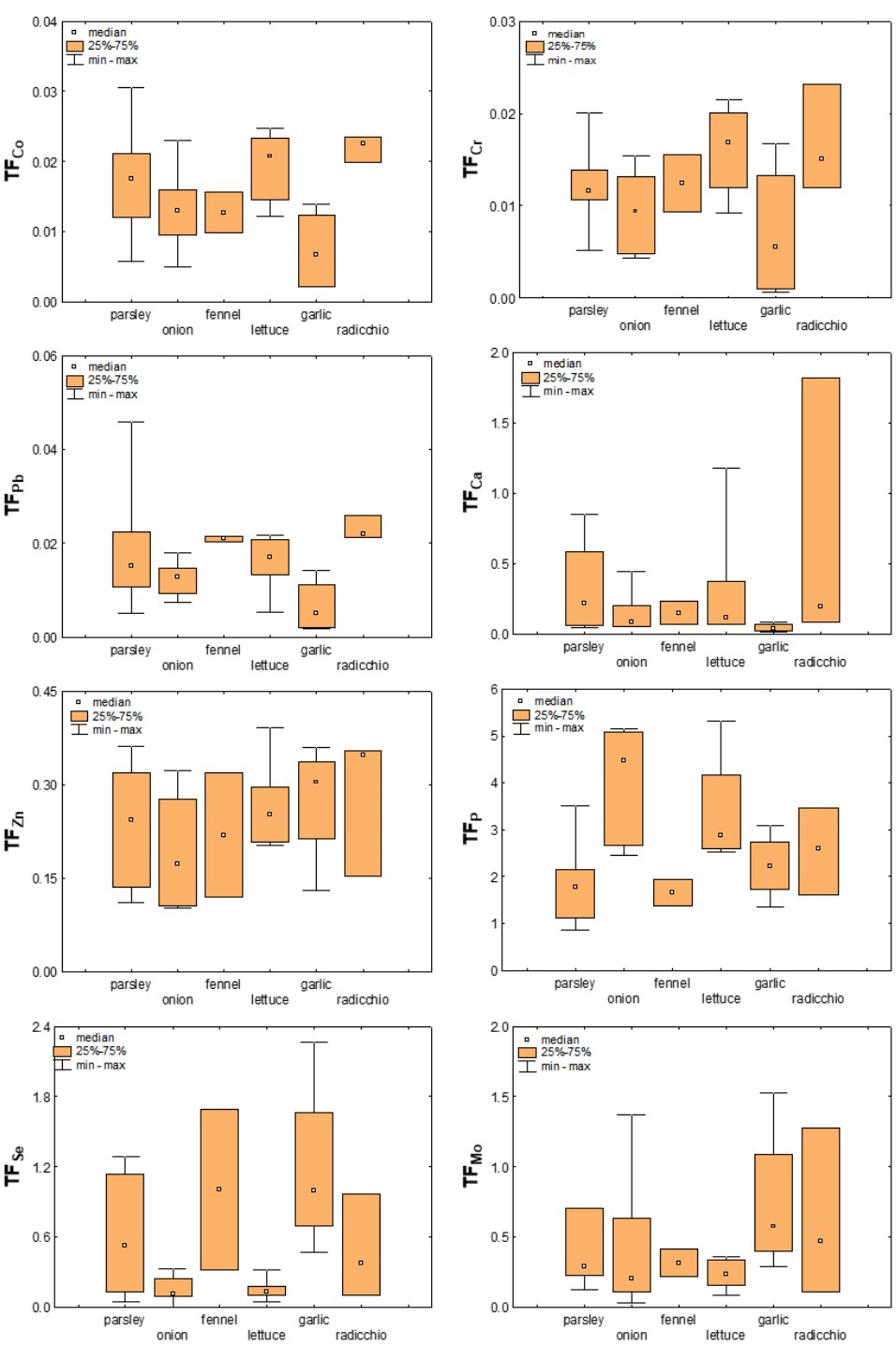

**Figure 7 Box-plots showing transfer factors of selected elements in different plant species.**

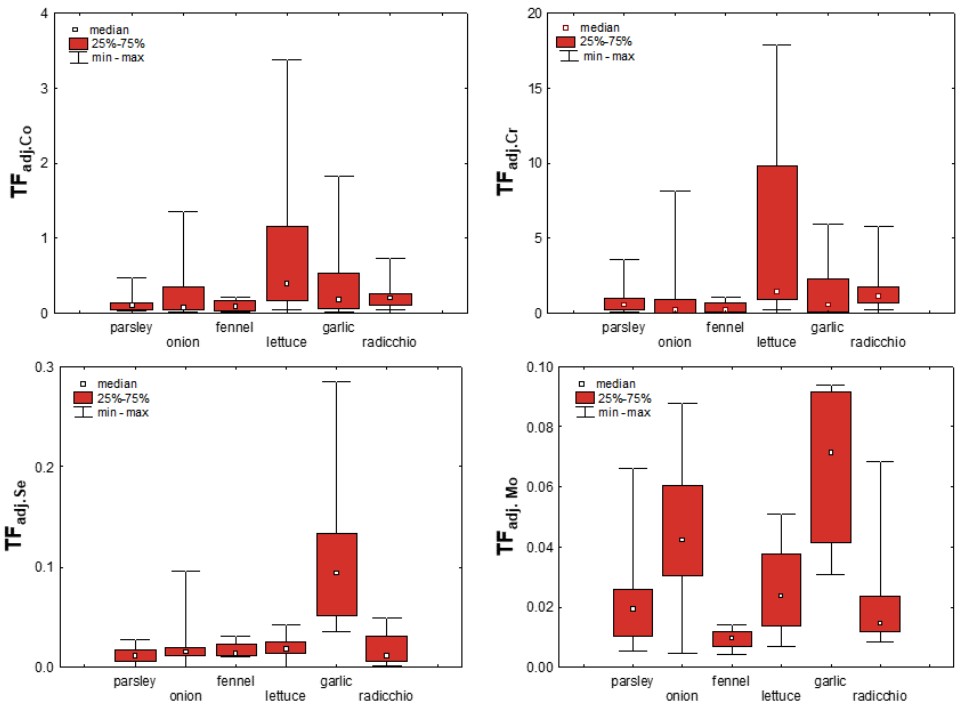

**Figure 8** Box-plots showing adjusted transfer factors of selected elements in different plant species.

metal uptake in terms of substrate properties and bioavailable element content, and even highlight to some extent the influence of plant species on element uptake.

## Influence of anthropogenic factors

A better understanding of the transfer of elements to edible plants is a prerequisite for sustainable soil management and the maintenance of soil quality and health, and all of this requires comprehensive data sets that include element data not only for vegetables but also for the associated soil and bioavailable fraction. However, the literature available in this regard is very limited, as authors study very different plant species using a very limited number of parameters (elements) and rarely provide data on both total metal content in plants and soils, and even more rarely on the bioavailable fraction in soil.

This is extremely important because each pollution source has its own metallic fingerprint that is reflected in the plants themselves. This is best illustrated in Figs. 9 and 10, where the distribution of TF in respectively, onions and in different vegetables grown on soils under the influence of different pollution sources, is shown. Figure 9 shows TF distribution data for Cd, Cr, Mn, and Pb in onions grown on nonpolluted soils in Iran (*Naghipour et al., 2018*), soils irrigated with wastewater and groundwater in Iran (*Cheshmazar et al., 2018*), or polluted river in Ethiopia (*Yirgalem & Alemnew, 2022*), or grown in areas under the influence of thermal power plants and combustion products in India (*Singh, Shikha & Saw, 2023*), including the study area (Supplementary material). It is evident from the Fig. 9 that the values are different for each element (Cd, Cr, Mn and

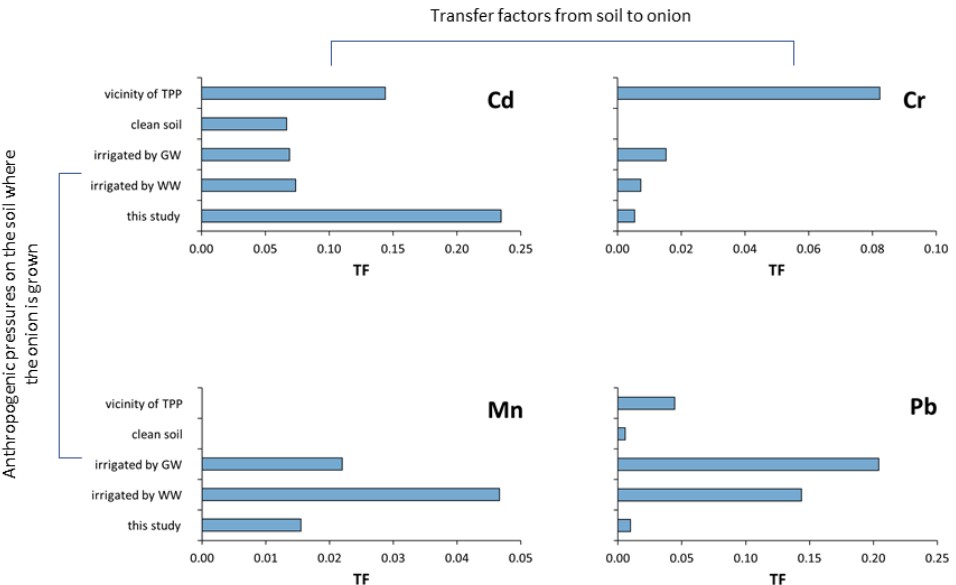

**Figure 9** Transfer factors for Cd, Cr, Mn, and Pb in onions growing on soils under different anthropogenic pressures.

Pb) and each pollution source. It is also obvious that even with similar pollution sources, the values of the mentioned metals are extremely different and do not show any regularity in the distribution. For example, the Cd content in the area of Raša and Opatija, where the influence of centuries-old mining and coal burning affects the content (*Medunić et al., 2016*), is significantly higher than the Cd content in Indian soil under the influence of TTP, while the opposite tendency applies to Cr and Pb.

Figure 10 shows a combined PCA plot by superimposing TF loadings and sample scores. For the above data set, the eigenvalues of the first two principal components (PCs) were greater than 1, indicating their importance. The first two PCs explained 73.3% of the total variability of the six variables; the first component (PC1) contributed 57.8% and the second (PC2) 15.5% to the total variance. From the above graph, it is clear that the plants from each area have a characteristic fingerprint that is reflected in the observed TFs. It can be seen that there is also an extremely large difference between the different regions, *i.e.*, pollution sources, with the samples from this study again standing out in terms of cadmium content.

Therefore, interpreting the contamination of vegetables with certain elements without insight into the nature of the substrate, its natural characteristics, as well as possible anthropogenic influences, may not only lead to incorrect conclusions, but also prevent the success of the management of these soils, as well as the implementation of measures necessary to maintain their quality.

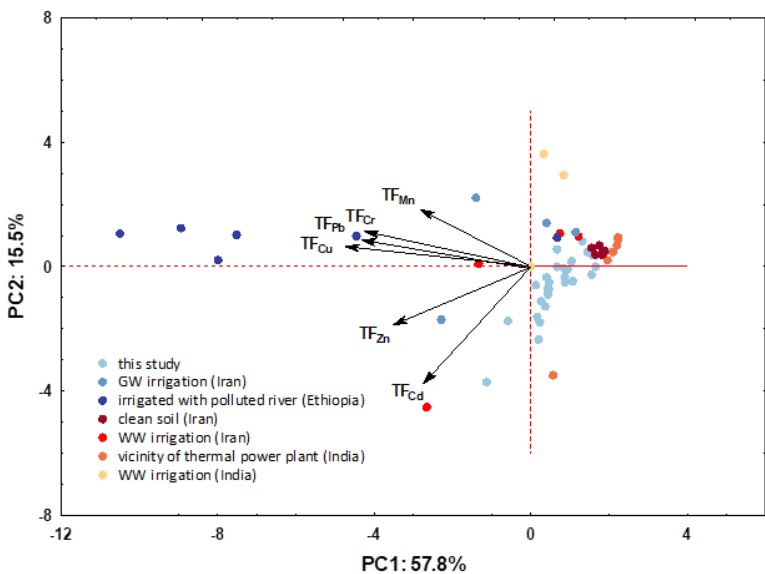

**Figure 10  PCA plot combining transfer factors loadings and soil scores for first two PCs.**

## CONCLUSIONS

The results clearly show that the multi-elemental composition of vegetables grown in the Raša and Opatija region is primarily determined by the naturally enriched soil substrate in combination with various anthropogenic pressures, especially the impact of mining and coal burning in the area over centuries. As a result, the metal(loid) content in some vegetables grown by locals in private gardens approaches the maximum allowable daily dose. In addition, the elements whose concentrations were directly affected by centuries of coal burning in the region (Mo, S, and Se) were found to not only have elevated levels in the vegetables, but also to affect the levels of other elements. This suggests a synergistic effect leading to higher than expected element levels in edible plant parts, including leaves and bulbs.

Moreover, since legislation does not fully cover these elements, their absence from the analysis may lead to misinterpretation of elevated concentrations of the monitored elements and inappropriate soil management actions. The data also reinforce the importance of assessing metal availability in specific environmental conditions using bioavailable concentrations of elements. Transfer factors based on bioavailable concentrations have proven to be a very useful tool, providing better insight into the uptake of elements as a function of their bioavailable content and the plant species itself.

## ACKNOWLEDGEMENTS

We would like to thank Mladen Bajramović for his help with sampling.

### Funding
This study has been supported by the Croatian Science Foundation under the project FORtIS (IP-2019-04-9354). The funders had no role in study design, data collection and analysis, decision to publish, or preparation of the manuscript.

### Grant Disclosures
The following grant information was disclosed by the authors:
Croatian Science Foundation under the project FORtIS: IP-2019-04-9354.

### Competing Interests
The authors declare that there are no competing interests.

### Author Contributions
- Marija Petrović performed the experiments, analyzed the data, authored or reviewed drafts of the article, and approved the final draft.
- Gordana Medunić conceived and designed the experiments, authored or reviewed drafts of the article, and approved the final draft.
- Željka Fiket conceived and designed the experiments, analyzed the data, prepared figures and/or tables, authored or reviewed drafts of the article, and approved the final draft.

### Field Study Permissions
The following information was supplied relating to field study approvals (i.e., approving body and any reference numbers):
Soil and vegetable samples were taken in private gardens with the prior consent of the owners.

### Data Deposition
The element concentrations in studied soils and vegetables, including transfer factors, are available in the Supplemental Files.

### Supplemental Information
Supplemental information for this article can be found online at http://dx.doi.org/10.7717/peerj.15904#supplemental-information.

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
