# Peer review of "Essential role of multi-element data in interpreting elevated element concentrations in areas impacted by both natural and anthropogenic influences"

_PeerJ, doi:10.7717/peerj.15904_

## Round 0.1 · original submission · Minor Revisions

Dear Dr. Fiket,
Please address the minor revisions identified by the reviewers.

Reviewer 1 ·

Basic reporting

The Manuscript focuses on the contamination of 27 metals out of which many are carcinogenic and have an adverse effect on ecological and human life. The author has referred to about 60 different references describing the outcome of the effect of metals on the environment.

All the facts, figures, and references included are as per the journal guidelines.

As a reviewer, I am able to read and understand the research intent and its effect on the environment.

Experimental design

The research activity is very well-defined and rigorous investigation is done.
In General, the experimental design was as per the research activity carried out.

There are some clarifications required
* One of your references which directs toward the thesis, includes 32 metal element studies on the
regions which you have mentioned in the article ( Page no -27 ). Also Why only 27 metals are
included in the manuscript if the research activity is carried out on 32? Request you include the
Human Health Risk Assessment parameter in the manuscript. This addition will strengthen the paper.
metals. Have gone through the research manuscripts, but this manuscript would give a researcher a
complete package on various metals and studies done.

Reference - Zorić T (2020)

https://www.academia.edu/79164673/Selenium_and_trace_metal_content_in_selected_vegetables_from_Ra%C5%A1a_and_Opatija_towns_and_their_estimated_daily_intakes
Also, the below paper Table-1 describes the Human health hazard.
https://www.frontiersin.org/articles/10.3389/fpls.2018.00732/full#B14 - da Rosa Couto et al., 2018

Some More gaps need to be filled in,

1. The analytical data is recorded using ICPMS - there are no chromatograms included in the
manuscript which gives strong support to the findings which are carried out during the analysis.
2. Page 20, Line 573 - Bakircioglu et el., 2011 - Research article refers to Wheat grain....this reference
is not suitable as we are looking for an effect on leafy vegetables.
3. Page 22, Line 654 - This reference could not be found in the research article. pl. check.
4. Page 24, Line 726 - The DOI mentioned in the literature is wrong, the correct one is DOI
10.1016/j.chemosphere.2018.02.008 , Kindly check and update.
5. Page 25, Line 761 - The DOI mentioned in the literature is wrong, the correct one is DOI:
10.1016/j.jenvman.2017.02.040 , Kindly check and update.
6. Page 26, Line 798 - Reference needs to be corrected. The correct pdf link is -
https://www.worldenergy.org/assets/images/imported/2016/10/World-Energy-Resources-Full-report-
2016.10.03.pdf - Kindly check and update.
7. Page 27, Line 828 - Kindly update the reference details as -
Environmental Chemistry, Published 2020
https://www.academia.edu/79164673/Selenium_and_trace_metal_content_in_selected_vegetables_from_Ra%C5%A1a_and_Opatija_towns_and_their_estimated_daily_intakes

These corrections would make this manuscript very strong.

Validity of the findings

The outcome of the results was comparable with the earlier work published.

Additional comments

This article is the only one which records so many metals. The thesis which is referred to has 32 metal elements but why 27 are reported doesn't know.

If Human health risk is also recorded in a tabulated form, then this paper would be one of the rarest and the best ones.

Reviewer 2 ·

Basic reporting

The research article is reliving the data generated for 27 metals by systematic study of soil and vegetable grown in the Raöa 530 and Opatija region to understand and demonstrate the effect of coal mining and combustions activity on human health.

The author has extended his study by calculating the transfer factor and trying correlations between soil and vegetable data. This work can inspire further investigation to correlate the effect of soil pollution on human health.

Experimental design

The experimental design, such as sample collection of soli and growing vegetables, further separation of vegetables into three parts, and their analysis to understand the levels of different metals, is well aligned with the discussion.
The transfer factors and statistical analysis are also aligned with the research work and give a reasonable correlation.

Scope to improve the manuscript
1. The manuscript is missing the specific instrumentation and chemistry section, which should have a consolidated list of instruments with their make and model in one place.
2. Similarly, the consolidated list of chemicals and the 27 metal reference standards used in the study with detail on make and purity should be in one place rather than distributed at varying places in the manuscript.
3. The detailed ICP-MS analytical method, as well as the limit of detection and linearity range for each metal, should be part of the manuscript.
4. Table 3: The Co and Cr element concentrations in eluates (mg/L) are mentioned as 0.0001 and < 0.0001. What is the limit of detection for these elements? Same observation for Pb, 0.00002, and 0.00002
5. Table 3: The test results are reported in different digits, for example, the co-element concentration in eluates (mg/L) is between 0.0001 and 0.00005. The author should follow the uniform digits and round figure methods for reporting the results. The reporting of 5-digit results is not adding value. The same observation is made for Pb results.
6. Page 22, Line 654: Reference is missing in manuscript

Validity of the findings

The overall study and its conclusion match the earlier work; additionally, the researcher tried to explain the transfer factor and how it has negative effects on human health.

Additional comments

The authors presented a highly descriptive approach to document the 27 metal results in soil and plants. However, it will be interesting to see this affect human or animal health and can give a high-level conclusion on the overall life cycle in future study.

---

## Round 0.2 · accepted · Accept

The authors followed all suggestions of the reviewer, making the paper suitable for publication.